# Application of Load-Sharing Concept to Mechanical Seals

**Mohsen Rahimpour [1], Alireza Samadani [2] and Saleh Akbarzadeh [2],***

1　Mechanical Engineering Group, Pardis Collage, Isfahan University of Technology, Isfahan 84156-83111, Iran
2　Department of Mechanical Engineering, Isfahan University of Technology, Isfahan 84156-83111, Iran
*　Correspondence: s.akbarzadeh@iut.ac.ir

**Abstract:** Mechanical seals are mechanisms that are used to prevent fluid leakage. Since the seal surfaces are in contact with one another, hydrodynamic and contact forces are functions of surface roughness. Additionally, since the lubrication regime under specific operating conditions such as low speed or high load causes the seal to operate in the mixed lubrication regime, thus the contact of asperities plays an important role. The primary purpose of this paper is to apply the load-sharing concept to study the behavior of a mechanical seal in a mixed lubrication regime. The predicted results are compared to the published data from the literature, showing acceptable accuracy. The model presented in this paper can predict the performance of the mechanical seal system in a short execution time while providing acceptable accuracy by considering the surface roughness effect.

**Keywords:** mechanical seals; asperities; load-sharing concept; mixed lubrication regime

## 1. Introduction

Mechanical seals are one of the most important parts of industrial machinery in oil and gas applications. They are used to prevent leakage from machines such as pumps, mixers, and agitators. Premature seal failure can have serious economic and environmental repercussions [1]. The performance of seals is one of the most important topics in tribology because it involves some of the most complex lubrication mechanisms. These mechanisms include the elastic deformation of surfaces, leakage, heat generation from viscosity loss, and changes in lubricant properties caused by variations in pressure, temperature, and film thickness [2].

In mechanical seals, the surfaces of the fixed part and the moving part are in contact with each other, and the corresponding contact prevents the fluid from escaping from the seal to the outside environment. Surface roughness is one of the most influential factors in the tribology of mechanical seals. For this purpose, the load-sharing concept has been used [3]. Based on this concept, the total load in the mixed lubrication regime is carried by the contact between asperities and the fluid film.

The mechanical seal has been the subject of numerous studies up to this point. In 1974, Mayer investigated the mechanical seal, the pertinent equations, and the effect of various parameters on the internal pressure of this seal [4].

Karaszkiewicz used experimental techniques in 1988 to investigate the distribution of pressure and film thickness. He calculated the leakage rate and assessed the effect of each parameter on the leakage using the film thickness distribution and data from the experimental approach [5].

Salant et al. proposed a controllable mechanical seal in 1989. His study of mechanical seals created mechanical seals that allow electrical sensors to regulate the film thickness between the fixed and moving parts. Compared to conventional seals, this technology enabled the fluid film thickness to be adjusted to an ideal level, decreasing wear between the components and improving the mechanical seal's useful life [6].

Salant performed a numerical analysis of the lip seal in 2001. He investigated three regimes of lubrication: hydrodynamic, elastohydrodynamic, and mixed lubrication. The

findings demonstrated that a mechanism provides the integrated fluid film's thickness. The main component of this mechanism, which stops fluid leakage, is a hydrodynamic force created by the trapped fluid between the surface roughness [7].

Key et al. (2004) used experimental methods to study the effects of various materials and fluids on a specific mechanical seal. According to his findings, silicone carbon is a reasonable choice for the mechanical seal's surface, and they proposed silicon carbon for the moving part [8].

Shen and Salant applied the mixed elastohydrodynamic lubrication model for radial seals in 2007. They demonstrated how the fluid film thickness distribution can perform better when using the elastohydrodynamic model. They also demonstrated how roughness on the shaft surface could help this seal perform better [9].

In 2013, Jia et al. used an elastohydrodynamic mixed lubrication model to study the effect of material and geometry on the radial lip seal. For this purpose, they employed the orthogonal array approach, an experimental technique with eight parameters that produced the desired parameter values at their best in five phases. The results for a radial lip seal with eight ideal parameters were superior to those of earlier seals in terms of performance, leakage rate, and friction torque [10].

In 2014, Rocke and Salant used flow factors to analyze a rotary lip seal in a mixed elastohydrodynamic lubrication regime. In this study, they considered the effect of surface roughness on flow factors such as elastic deformation and fluid properties. The collected findings demonstrated that the employed method is up to 30 times faster than statistical methods. They also demonstrated how this approach could be used to predict parameters such as fluid film thickness, power loss, and other necessary characteristics and produce accurate results [11].

Zhang et al. conducted a numerical analysis of the mechanical seal while considering the effect of contact force. They analyzed the deformation and pressure distribution of the moving and fixed parts while using Ansys software to carry out the numerical simulation method. They demonstrated that the contact width reduces with increasing rotational speed, and the deformation of the fixed and moving parts increases with increasing spring force [12].

Martsynkowsky et al. investigated the mechanical seal's dynamic behavior in 2015 and studied a few potential failure factors. For this purpose, they considered the axial vibration under the influence of the period for the seal's dynamic analysis. They concluded that the axial vibration and the generated frequencies are one potential failure factor, and the number of frequencies and oscillations depends on the kinetic energy produced by the axis [13].

Liu et al. investigated a wavy-tilt-dam mechanical seal by considering a three-dimensional model of thermal electrohydrodynamics in different working conditions. The results showed that the performance of the seal changes with increasing pressure and rotation speed [14].

Miguot et al. conducted a numerical analysis to determine how temperature changes affect the performance of a mechanical seal that uses water to operate. The collected data demonstrated that as the force increases, the temperature rises, increasing deformation and heat transfer. They compared the numerical analysis results with the experimental data, which showed an acceptable agreement [15].

Liu et al. developed a model to predict wear in the mixed-lubrication regime in mechanical seals using flow factors, and a parametric study was also presented [16]. Salant et al. modeled the performance of the hydraulic cylinder in a reactor. His main goal was to decrease RCP (reactor cooling pump) seal leakage. This seal has a non-rotating ring plate pressurized with a fluid, causing deformation. They demonstrated that the fluid film thickness and quantity of leakage between the sealing surfaces could be controlled based on the plate's geometry [17].

In 2018, Cochain conducted a numerical and experimental study on the seal under actual pressure. In this study, he showed that increased surface roughness increases fluid

film thickness and leakage [18]. In 2019, Li et al. studied friction's effect on mechanical seals' performance using the finite element method [19]. The load-sharing method is common in modeling contacts in the mixed lubrication regime. The main idea of this method is that during the contact of two surfaces in the mixed lubrication regime, a fraction of the load is carried by the asperities of the surface, and the lubricating film carries the rest.

In 2022, Yin et al. investigated a reciprocating piston seal using the fractional state transfer (FST) method. They studied the friction caused by heat sources originating from reciprocating movement in the mixed lubrication regime [20]. Cheng et al. investigated a multi-lip reciprocating seal in the steady state mixed lubrication regime. They studied contact pressure, leakage, friction coefficient, and seal performance and applied the Greenwood–Williamson approach to examine the contact pressure. They used the Patir–Cheng method to select the flow parameters in order to solve the modified Reynolds equation [21].

Nomikos et al. investigated the leakage of edge seals using the analytical-experimental method. They investigated the effects of surface roughness on leakage and used Patir–Cheng relationships to determine the flow factors [22]. They demonstrated that surface roughness affects leakage [23].

This paper presents the results of a numerical investigation examining surface roughness's effect on mechanical seal performance. It explores the impact of surface roughness on key factors such as film thickness distribution, hydrodynamic pressure distribution, and contact pressure. Additionally, various parameters, including fluid viscosity, speed, and applied force, were considered in the analysis. For this purpose, the load-sharing method has been employed. In order to predict the film thickness and friction coefficient in the mixed lubrication regime, the common equations in the full-film lubrication regime have been modified [24]. The load-sharing method offers the advantage of simultaneously considering various parameters, including fluid properties, sliding speed, surface roughness, friction coefficient, temperature effects, and other effective factors. This combined solution of the equations enables a comprehensive examination of their effects on the system.

In 1971, Johansson et al. investigated the effect of surface roughness between components in the mixed lubrication regime [25].

By considering the surface roughness, Masjedi and Khonsari examined the fluid film thickness, the contact force in the mixed lubrication regime, and line contact between the surfaces [26]. Then, they numerically investigated the friction coefficient in mixed lubrication conditions [27].

In the mixed lubrication regime, Masjedi and Khonsari investigated numerically and experimentally the friction coefficient between rough surfaces in the line contact [28]. In 2023, Fatourehchi et al. studied the performance of mechanical seals in hydrostatic and hydrodynamic conditions using the developed method. They demonstrated the efficacy of the developed approach for predicting seal performance and determining leakage. However, it should be noted that it is difficult and time-consuming to solve the Reynolds equation in the mixed lubrication regime [29].

The load-sharing method allows for the simultaneous solution of multiple equations while being more practical, accurate, and efficient. Various techniques, including simulation methods and experimental and numerical tests, have been conducted to evaluate the performance of mechanical seals in different working conditions and for all types of mechanical seals. However, each method has a long execution time and is extremely complicated.

In this study, the performance of a mechanical seal has been investigated using the load-sharing method. In this method developed for the contact in the mixed lubrication regime, the fluid film and the asperities contact carry the total applied load. The application of the load-sharing method to the contact of rough surfaces such as gears, cam followers, etc. has revealed that this method is capable of predicting the performance of tribo-systems with acceptable accuracy and a short execution time.

## 2. Simulation

The fixed and moving parts of the mechanical seal are in contact with each other, as described in the Introduction section. Figure 1 depicts the intersection of two contact plates. The contact created is linear, and the mixed lubrication regime is considered according to the roughness of the surfaces and fluid penetration in the space between them [30].

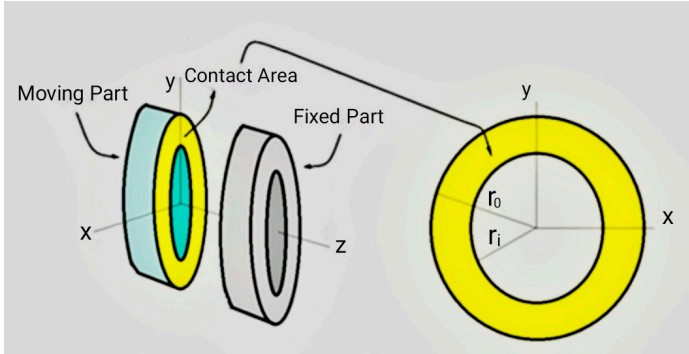

**Figure 1.** Intersection of the surfaces of fixed and moving parts.

In this regime, a thin lubricant layer known as the lubrication film carries most of the applied load. The contact of asperities also contributes to carrying the total load, which raises the surface temperature and causes wear, and on the other hand, changes the topography of the surface due to the plastic or elastoplastic deformation of asperities [29].

Regardless of the effect of surface roughness, contact analysis of mechanical seal surfaces requires the simultaneous solving of Reynolds equations and elastic surface deformation for large numbers of points inside the contact zone. This process takes a significant amount of time. When surface roughness is considered, the problem becomes more complicated, and the modified Reynolds equation must be used to calculate lubricant pressure changes [30].

The load-sharing approach makes it possible to complete this process in a very short period of time and with significant accuracy. Equation 1 is the fundamental equation in this approach. This equation divides the overall pressure into two parts: hydrodynamic and contact pressure. The total pressure, $P$, can be written as the sum [31]:

$$P = P_d + P_f \tag{1}$$

where $P_f$ is the hydrodynamic pressure and $P_d$ represents the pressure carried by asperities (Figure 2) [31]. In order to obtain the hydrodynamic pressure distribution, the Reynolds equation is used, and solving this equation is complicated and difficult. On the other hand, the ZMC model is considered due to the fact that wear is usually produced uniformly in the contact area; for this purpose, the desired contact is defined as a linear contact. The ZMC equation is applied in order to analyze the contact pressure distribution. Zhao and colleagues presented an elastic-plastic model known as ZMC for the contact of surface asperities. The long transition from elastic deformation to fully plastic deformation is the main characteristic of this model. The results of contact analyses of asperities show that the elastoplastic contact of asperities plays an important role in the contact behavior of rough surfaces [32]. Beheshti and Khonsari showed that the ZMC model provides the closest results compared to other models [33].

In elastohydrodynamic contact, surfaces are separated by thin layers of lubrication that have a lower level of thermal conductivity. The viscosity of the lubrication in these contacts changes from a low value at the inlet to a maximum value at the center and decreases again at the outlet. At the same time, the pressure on the lubricant increases as it is drawn to the contact surface, which causes a change in viscosity. Therefore, the force required to cut the lubricant layer changes in the contact zone, and more heat is produced in the areas where the viscosity is the highest. This correlation between the viscosity and the heat produced

affects the temperature distribution. Considering that the thermal conductivity coefficient ($K$) is constant, the energy equation is expressed as Equation (2) [34].

$$\rho C_p \left[ u_x \frac{\partial T}{\partial x} + u_y \frac{\partial T}{\partial y} + u_z \frac{\partial T}{\partial z} \right] + \frac{T}{\rho} \frac{\partial \rho}{\partial T} (u_x \frac{\partial p}{\partial x} + u_y \frac{\partial p}{\partial y} + u_z \frac{\partial p}{\partial z}) \left[ \frac{\partial}{\partial x} \left( K \frac{\partial T}{\partial x} \right) + \frac{\partial}{\partial y} \left( K \frac{\partial T}{\partial y} \right) + \frac{\partial}{\partial z} \left( K \frac{\partial T}{\partial z} \right) \right] + \eta [(\frac{\partial u_x}{\partial z})^2 + (\frac{\partial u_y}{\partial z})^2] + \frac{c p_d u_s}{h} \tag{2}$$

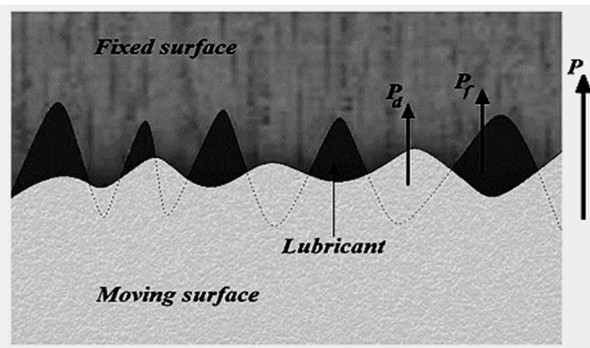

**Figure 2.** Hydrodynamic and contact forces between surfaces.

In deriving the governing equations in this problem, it is considered that surfaces are smooth without any roughness. However, it is not a particularly realistic assumption due to the fact that the surfaces actually have roughness. Furthermore, the working conditions of the contact between the two surfaces may be such that the film thickness of the lubricant formed between the two surfaces can be smaller than the average roughness of the surfaces. In this case, the effect of roughness cannot be ignored, and a mixed lubrication regime is established. In this case, the governing equations of hydrodynamic forces need to be modified for these concepts. Equation (3) indicates the modified Reynolds equation for the hydrodynamic forces of rough surfaces [9]:

$$\frac{\partial}{\partial x} \left( \varnothing_x \frac{\partial p_f}{\partial x} \right) + \frac{\partial}{\partial y} \left( \varnothing_y \frac{h^3}{\eta} \frac{\partial p_f}{\partial y} \right) = \frac{U_1 + U_2}{2} \frac{\partial \overline{h}_T}{\partial x} + \frac{U_1 - U_2}{2} \sigma \frac{\partial \varnothing_s}{\partial y} + \frac{\partial \overline{h}_T}{\partial t} \tag{3}$$

where $\varnothing_x$, $\varnothing_y$, and $\varnothing_s$ are fluid film pressure and indicators of the average flow ratio in the contact of the rough surfaces. It is very common to consider the roughness to be the same for both surfaces with an isotropic pattern; therefore, such an assumption is also considered in this study. Where it is impossible to use a statistical roughness distribution, it is necessary to obtain an accurate and definite surface roughness distribution by measuring it. The flow factors are based on the Patir-Cheng approach to analyze and solve the modified Reynolds equation [22]. The standard form of the Reynolds equation is used for the elastohydrodynamic lubrication regime between two surfaces. Instead, the film thickness equation of lubrication is below for the line contact. The basis for studying the desired contact between the asperities as a linear contact is that the contact surfaces in mechanical seals make relatively uniform contact with one another. In this equation, $\acute{\delta}$ represents the elastic deformation and $\delta_1$ represents the roughness height [24].

$$h(x) = h_0 + \frac{x^2}{2R_x} + \delta_1(x) + \acute{\delta}(x) \tag{4}$$

Another effect of roughness is the change in load balance. In such cases, the load is shared between film and asperities, and it is necessary to modify the load equation as Equation (5) [34].

$$w = \iint p \, dA = \iint \left( p_f + p_c \right) dA \tag{5}$$

For the contact part, three equations are achieved to predict the central film thickness, the minimum film thickness, and the load intensity ratio in line contact with the surface roughness condition. These equations are based on the simultaneous solution of the modified Reynolds equation and surface deformation according to surface roughness in elastic, plastic, and elastoplastic deformation. The equations cover a wide range of input, and they are $f(W, U, G, \bar{\sigma}, V)$. Figure 3 shows the contact geometry of two fixed and moving parts [30].

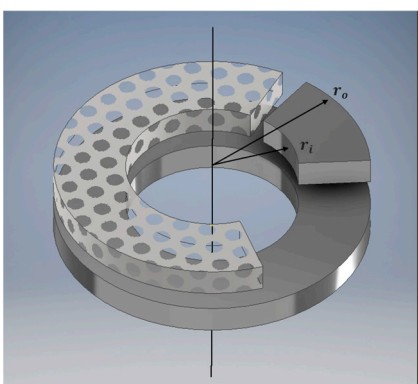

**Figure 3.** Contact geometry of fixed and moving components of mechanical seal.

Figure 4 shows the effect of contact on surface asperity:

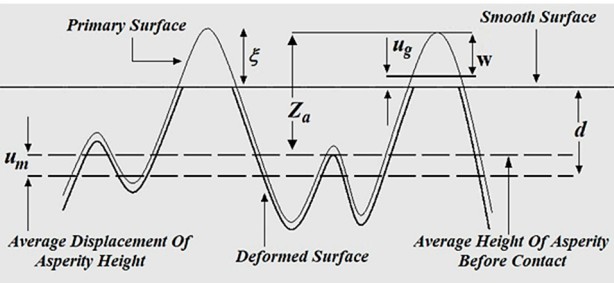

**Figure 4.** Contact of an asperity with a rigid and polished surface.

When the interaction of surface asperities is considered, some changes in the overall surface roughness occur. As shown in Figure 4, the average displacement of the asperity height is $u_m$, $d$ represents the average surface distance of two surfaces, $Z_a$ represents the height of a specific asperity before contact with another surface, and $\xi$ is the overall interference or height of the mentioned asperity after contact. If $u_g$ is considered the deformation caused by pressure on another asperity in contact, the following geometric relationship can be deduced from Figure 4 [35]:

$$w = Z_a - d + \left(u_m - u_g\right) \tag{6}$$

where quantity $u_m - u_g$ shows the influence of asperity interference on the local deformation behavior of a specific asperity. This quantity should be a function of the surface roughness, total average pressure, and material parameters. On the other hand, it can be assumed that any asperity during contact has an area or territory on the surface around it, and the higher the asperity, the bigger the territory. This assumption can be represented with a relative equation between the surface region, $A_i$, and the load applied by the asperity $W_i$ [27]:

$$A_i = \lambda W_i \tag{7}$$

The sum of all regions of the asperities achieves the contact area $A_i$, which can be expressed as $A_n = \sum A_i$. On the other hand, the coefficient $\lambda$ is:

$$\lambda = \frac{A_n}{\sqrt{W_i}} = \frac{A_n}{W_t} = \frac{1}{p_m} \tag{8}$$

On the other hand, the average contact pressure is:

$$p_m = \frac{W_t}{A_n} = \frac{W_i}{A_i} \tag{9}$$

Based on this equation, it can be concluded that the asperity pressure throughout the asperity region is equal to the overall contact pressure $p_m$. If $N_a$ is considered the number of asperities on the surface with a nominal area $A_n$, then the contact number of asperities can be achieved using the following relationship [29]:

$$\text{The number of contacts between asperities} = N_a = \int_d^\infty \phi(z)dz = n_a A_n \int_d^\infty \phi(z)dz \tag{10}$$

where $n_a$ is the surface density of asperities and $\phi(z)$ is the distribution density of the height of asperities. $A_t$ is the overall actual contact surface, and total load, $w_t$, is achieved from the total distribution of each of the asperities in all its small contacts. Therefore, for a certain distance between two surfaces, the contact surface and the total actual load are obtained from the following equation [29]:

$$\begin{aligned}
A_t &= A_e + A_p + A_{ep} \\
&= N\left(\int_d^{d+\omega_1} A_e\phi(z)dz + \int_{d+\omega_2}^\infty A_p\phi(z)dz + \int_{d+\omega_1}^{d+\omega_2} A_{ep}\phi(z)dz\right) \\
&= n_a A_n \pi R \int_d^{d+\omega_1} \omega\phi(z)dz + 2n_a A_n \pi R \int_{d+\omega_2}^\infty \omega\phi(z)dz \\
&+ n_a A_n \pi R \int_{d+\omega_1}^{d+\omega_2} \omega\left[1 - 2(\frac{\omega-\omega_1}{\omega_2-\omega_1})^3 + 3(\frac{\omega-\omega_1}{\omega_2-\omega_1})^2\right]\phi(z)dz
\end{aligned} \tag{11}$$

$$\begin{aligned}
w_t &= w_e + w_p + w_{ep} \\
&= N\left(\int_d^{d+\omega_1} w_e\phi(z)dz + \int_{d+\omega_2}^\infty w_p\phi(z)dz + \int_{d+\omega_1}^{d+\omega_2} w_{ep}\phi(z)dz\right) \\
&= \frac{4}{3}n_a A_n E R^{0.5} \int_d^{d+\omega_1} \omega^{1.5}\phi(z)dz + 2n_a A_n \pi R(Hd)\int_{d+\omega_2}^\infty \omega\phi(z)dz \\
&+ \left\{n_a A_n \pi R \int_{d+\omega_1}^{d+\omega_2} \omega\left[(Hd) - 0.6Hd\left(\frac{Ln\frac{\omega_2}{\omega}}{Ln\frac{\omega_2}{\omega_1}}\right)\right]\right. \\
&\left.\times \left[1 - 2(\frac{\omega-\omega_1}{\omega_2-\omega_1})^3 + 3(\frac{\omega-\omega_1}{\omega_2-\omega_1})^2\right]\phi(z)dz\right\}
\end{aligned} \tag{12}$$

These two equations can be dimensionless, respectively, by dividing by $A_n$ and $A_n E$. Additionally, the longitudinal parameters of these equations, such as $\omega, \omega_2, \omega_1, y_s$, and $z$ can be dimensionless by dividing by the standard deviation of the surface roughness distribution $\sigma$ and with the superscript star, respectively, and can be shown in the form $h^*, \omega^*, \omega_2^*, \omega_1^*, y_s^*$, and $z^*$. Then, the final dimensionless equations can be written as [27]:

$$\begin{aligned}
A_t^* &= \frac{A_t}{A_n} = \pi\beta\int_{h^*-y^*}^{h^*-y^*+\omega_1^*} \omega^*\phi^*(z^*)dz^* + 2\pi\beta\int_{h^*-y^*+\omega_2^*}^\infty \omega^*\phi^*(z^*)dz^* \\
&+ \pi\beta\int_{h^*-y^*+\omega_1^*}^{h^*-y^*+\omega_2^*} \omega^*\left[1 - 2(\frac{\omega^*-\omega_1^*}{\omega_2^*-\omega_1^*})^3 + 3(\frac{\omega^*-\omega_1^*}{\omega_2^*-\omega_1^*})^2\right]\phi^*(z^*)dz^*
\end{aligned} \tag{13}$$

$$\begin{aligned}
W_t^* &= \frac{W_t}{A_n E} = \frac{4}{3}\beta(\frac{\sigma}{R})^{0.5}\int_{h^*-y^*}^{h^*-y^*+\omega_1^*} \omega^{*1.5}\phi^*(z^*)dz^* \\
&+ \frac{2\pi\beta(Hd)}{E}\int_{h^*-y^*+\omega_2^*}^\infty \omega^*\phi^*(z^*)dz^* \\
&+ \left\{\frac{\pi\beta(Hd)}{E}\int_{h^*-y^*+\omega_1^*}^{h^*-y^*+\omega_2^*} \omega^*\left[1 - 0.6(\frac{Ln\frac{\omega_2^*}{\omega}}{Ln\frac{\omega_2^*}{\omega_1^*}})\right] \times \left[1 - 2(\frac{\omega^*-\omega_1^*}{\omega_2^*-\omega_1^*})^3 + 3(\frac{\omega^*-\omega_1^*}{\omega_2^*-\omega_1^*})^2\right]\phi^*(z^*)dz^*\right\}
\end{aligned} \tag{14}$$

where $\beta = n_a$ and $\omega^* = z^* - h^* + y_s^*$ and $\phi^*(z^*)$ can be considered as follows:

$$\phi^*(z^*) = \frac{1}{(2\pi)^{0.5}}\left(\frac{\sigma}{R}\right)\exp\left[-0.5(\frac{\sigma}{\sigma_s})^2 z^{*2}\right] \tag{15}$$

It should be noted that in a numerical approach, the film thickness $h$ is equal to the separation, which is the distance between the average lines of two rough surfaces. In Equation (16), the roughness radius is $\beta$; $n$ represents the roughness density coefficient; $\sigma_s$ is an indicator of the peak asperities' standard deviation; and $y_s$ indicates the average distance between the height line and the asperities' peak line. Based on McCall's conclusions, these two parameters can be written as [27].

$$y_s = \frac{0.0459}{n\beta\sigma}. \qquad \sigma_s = \sqrt{1 - \frac{3.7169 \times 10^{-4}}{(n\beta\sigma)^2}}\sigma \tag{16}$$

Equation (17) describes the contact pressure distribution based on the ZMC model.

$$
\begin{aligned}
p_d &= \frac{2}{3}E'n\beta^{0.5}\alpha^{1.5}\left(\frac{\sigma}{\sigma_s}\right)\frac{1}{\sqrt{2\pi}}\int_{h^*-y_s^*}^{h^*-y_s^*+w_1^*} w^{*1.5} e^{-0.5(\frac{\sigma}{\sigma_s}Z^*)^2}dZ^* \\
&+2\pi hdn\beta\sigma\left(\frac{\sigma}{\sigma_s}\right)\frac{1}{\sqrt{2\pi}}\int_{h^*-y_s^*+w_2^*}^{\infty} w^* e^{-0.5(\frac{\sigma}{\sigma_s}Z^*)^2}dZ^* \\
&+\pi hdn\beta\sigma\left(\frac{\sigma}{\sigma_s}\right)\frac{1}{\sqrt{2\pi}}\int_{h^*-y_s^*+w_1^*}^{h^*-y_s^*+w_2^*} w^* e^{-0.5(\frac{\sigma}{\sigma_s}Z^*)^2} \\
&\times\left[1-0.6\frac{Lnw_2^*-Lnw^*}{Lnw_2^*-Lnw_1^*}\right]\times\left[1-2(\frac{w^*-w_1^*}{w_2^*-w_1^*})^3\right. \\
&\left.+3(\frac{w^*-w_1^*}{w_2^*-w_1^*})^2\right]dZ^*
\end{aligned} \tag{17}
$$

In order to study the surface asperities and the effect of the contact force, the ZMC model described in Equation (17) is used. Several inputs for the surface texture are needed, but some of them can be removed with a little compromise. $n\beta$ parameter does not differ much for different surfaces, and it can be considered an almost constant value. Many researchers have assumed this parameter to be 0.05 in their articles, based on Equation (16), $y_s = 0.92\sigma$ and $\sigma_s = 0.92\sigma$ [27].

Therefore, according to these assumptions, the inputs of the hydrodynamic equation for rough surfaces are load, velocity, lubricant, the radius of asperity, the standard deviation of surface height, and the surface curve. Equation (17) expresses the pressure distribution on the asperities. The integral in Equation (17) can be calculated numerically with appropriate accuracy using the Gauss–Legendre method, considering five points.

Another assumption that can be used in the study is to consider $R$ as the radius of the asperity equal to 0.01 of the combined standard deviation of surface roughness. In order to make the ZMC equation dimensionless, the coefficients are considered Equation (18) [27].

$$
\begin{aligned}
&\bar{\beta} = \frac{\beta}{R}, \bar{n} = nR^2, V = \frac{hd}{E}, \bar{\sigma}_s = \frac{\sigma_s}{R}, \bar{y}_s = \frac{y_s}{R} \\
&\bar{w}_1 = (0.6\pi V)^2\bar{\beta}, I_1 = \frac{H-\bar{y}_s}{\bar{\sigma}}, I_2 = \frac{H-\bar{y}_s+\bar{w}_1}{\bar{\sigma}}, I_3 = \frac{H-\bar{y}_s+\bar{w}_2}{\bar{\sigma}}
\end{aligned} \tag{18}
$$

In this case, Equation (17) is obtained as Equation (19):

$$
\begin{aligned}
p_d = {} & \tfrac{2}{3}\bar{n}\bar{\beta}^{-0/5}\bar{\sigma}^{1/5}W^{-0/5}\left(\frac{\bar{\sigma}}{\bar{\sigma_s}}\right)\frac{1}{\sqrt{2\pi}}\int_{I_1}^{I_2}(Z^*-I_1)^{1/5}e^{-0/5\left(\frac{\bar{\sigma}}{\bar{\sigma_s}}Z^*\right)^2}dZ^* \\
& +2\pi V\bar{n}\bar{\beta}\bar{\sigma}W^{-0/5}\left(\frac{\bar{\sigma}}{\bar{\sigma_s}}\right)\int_{I_3}^{\infty}(Z^*-I_1)e^{-0/5\left(\frac{\bar{\sigma}}{\bar{\sigma_s}}Z^*\right)^2}dZ^* \\
& +\pi V\bar{n}\bar{\beta}\bar{\sigma}W^{-0/5}\left(\frac{\bar{\sigma}}{\bar{\sigma_s}}\right)\frac{1}{\sqrt{2\pi}}\int_{I_2}^{I_3}(Z^* \\
& -I_1)e^{-0/5\left(\frac{\bar{\sigma}}{\bar{\sigma_s}}Z^*\right)^2}\times\left[1-0/6\frac{Ln\bar{w}_2-Ln(Z^*-I_1)}{Ln\bar{w}_2-Ln\bar{w}_1}\right]\times[1 \\
& -2(\frac{(Z^*-I_1)-\bar{w}_1}{\bar{w}_2-\bar{w}_1})^3+3(\frac{(Z^*-I_1)-\bar{w}_1}{\bar{w}_2-\bar{w}_1})^2]dZ^*
\end{aligned}
\tag{19}
$$

Therefore, the input parameters for the elastohydrodynamic problem are $W, U, G, \bar{\sigma}, \bar{\beta}$, and $V$.

Figure 5 is the contact force calculation algorithm for temperature mode. In this case, the goal is to achieve the load-sharing equation.

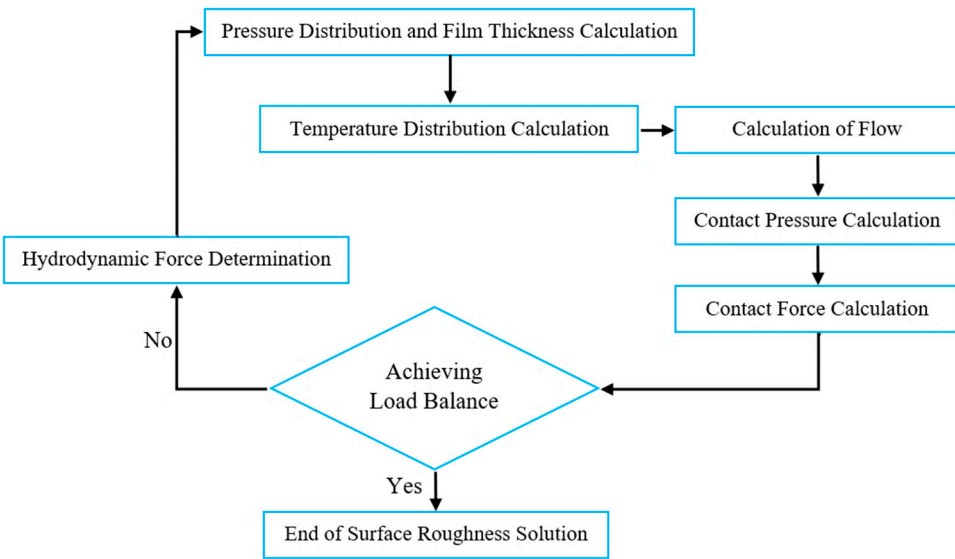

**Figure 5.** Contact pressure distribution algorithm.

It is worth noting that the mechanical seal force analysis can be viewed from a different point of view. Figure 6 shows how different forces are applied to the sealing components. These forces include the opening force ($W$), hydrodynamic force ($W_h$), spring force ($W_s$) and friction force ($W_f$).

Most seals operate with some combination of hydrodynamic and mechanical contact, as has been discussed in this paper. The hydrodynamic and mechanical contact forces can be presented as one term, $P_f A_f$.

$$
F_o = P_2\left(A_c - A_f\right) + P_f A_f + 2\pi\int_{R_2}^{R_1}Prdr
\tag{20}
$$

The closing force, $F_c$, has two components, as shown in Equation (21). The first component is the sealed pressure, $P_1$, which acts on the area at the end of the seal, represented by $A_c$, resulting in a closing force of $P_1 A_c$. Additionally, for most seals, there is a mechanical

spring force, denoted as $F_{sp}$, which also acts on the area $A_c$, providing an initial closing force.

$$F_c = P_1 A_c + F_{sp} \tag{21}$$

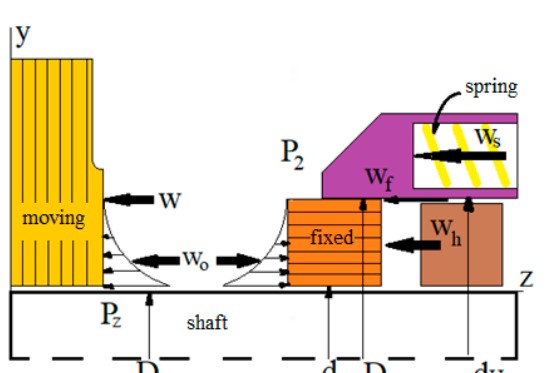

**Figure 6.** Applying the forces on the sealing component.

For steady state operation, the opening and closing forces are equal. Therefore, the average face pressure, $P_f$, may be found by equating Equations (20) and (21).

$$P_f = P_1 \left( \frac{A_c}{A_f} \right) - P_2 \left( \frac{A_c}{A_f} - 1 \right) - \frac{2\pi}{A_f} \int_{R_2}^{R_1} Pr dr + \frac{F_{sp}}{A_f} \tag{22}$$

If $P_f$ is positive, there may be direct rubbing between the moving and fixed faces; this is a condition of low leakage. For $P_f$ to be negative, the opening forces must overcome the closing forces, and leakage may be high. Equation (22) is cumbersome to use but is basic to an understanding of mechanical seal design.

Another important parameter in mechanical seals is the load factor, which represents the relationship between the forces acting on the seal but is basically a geometric ratio. The load factor is equal to the ratio of the surface that is exposed to hydrodynamic pressure to the contact surface of the sealing surfaces, as illustrated in Figure 7, and is defined as:

$$\frac{A_h}{A_f} = \frac{(D_b^2 - D_i^2)}{(D_o^2 - D_i^2)} \tag{23}$$

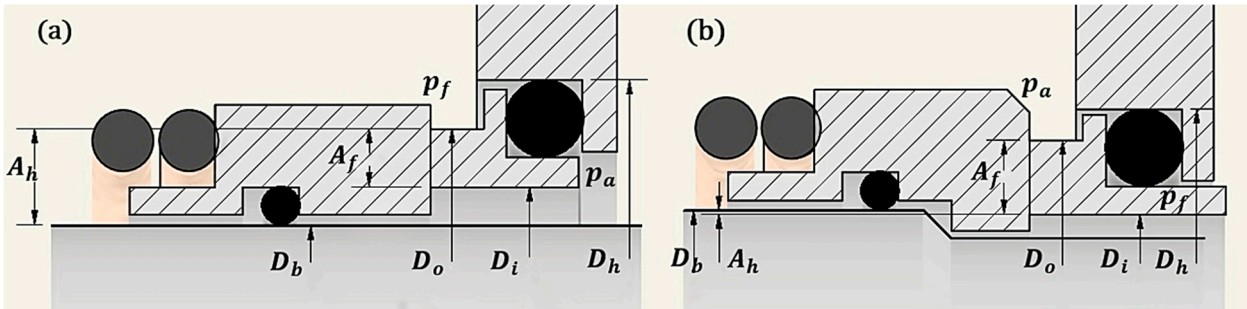

**Figure 7.** (**a**) Unbalanced outside pressurized seal, internally mounted, (**b**) balanced inside pressurized seal, externally mounted.

The load factor for this paper's mechanical seal is between 0.8 and 1.2.

## 3. Results

The hydrodynamic and contact pressures have been separated using the load-sharing method and considering the pressure distribution. Furthermore, the effect of surface roughness on the pressure distribution has been investigated for several surface samples

with different roughnesses for the contact of the seal components. Detailed figures have been generated for each component, illustrating the pressure distribution and highlighting the impact of surface roughness. Figure 8 shows the film thickness distribution for 0.5 (μm) surface roughness. It should be noted that the results were validated based on reference [35] and the parameters of Table 1 at a pressure of 3 MPa.

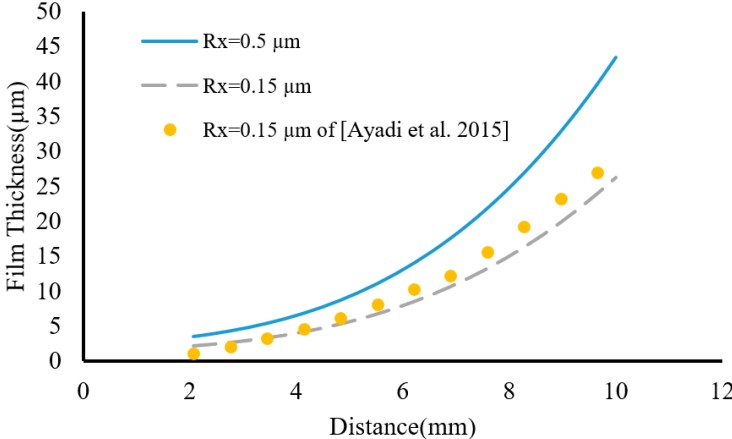

**Figure 8.** Film thickness distribution on the surface with a roughness of 0.15 (μm) [35].

**Table 1.** Input parameters based on reference [35].

| Parameters | Unit | Value | |
|---|---|---|---|
| Dry friction coefficient | | 0.15 | |
| Viscosity | mPa s | 0.65 | |
| Flow rate | 1/min | 30 | |
| | | Stator | Rotor |
| Young's modulus | GPa | 33.35 | 393 |
| Passion's coefficient | | 0.28 | 0.2 |
| Conductivity | W/m K | 13 | 125 |
| Expansion coefficient | 1/K | $4.5 \times 10^{-6}$ | $4.3 \times 10^{-6}$ |
| Roughness | μm | 0.1 | 0.15 |

Figure 8 shows the film thickness distribution on the surface with a roughness of 0.15 (μm). The results have been validated based on Ayadi and colleagues' findings [35].

The results achieved from the film thickness distribution along the contact surfaces for rough surfaces show that the film thickness increases from the inner to the outer diameter. Specifically, it has the highest value in the vicinity of the outer diameter. The comparison of the graph for the rough surfaces shows that the roughness of the surfaces leads to a change in the slope of the graph. As the surface roughness increases, the slope of the film thickness variation increases.

The roughness of the desired surface in this section is defined in two ranges: a minimum roughness of 0.15 (μm) and a maximum roughness of 0.5 (μm).

The contact force in relation to surface roughness is depicted in Figure 9. It is demonstrated that the contact force increases as the roughness of the contact surface increases.

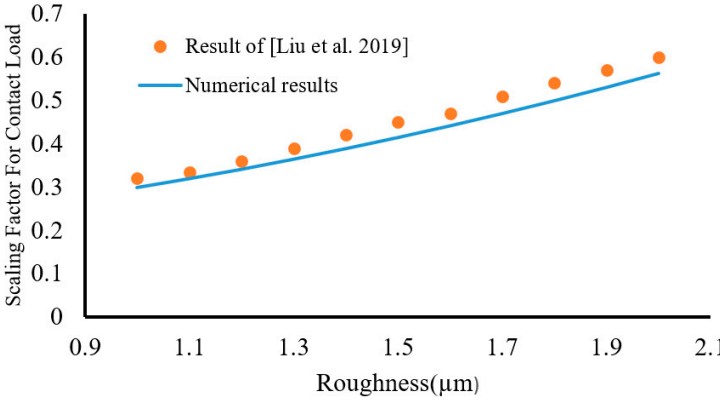

**Figure 9.** Contact load relative to the surface roughness [16].

On the other hand, Figure 10 shows the pressure distribution on the surface with higher roughness.

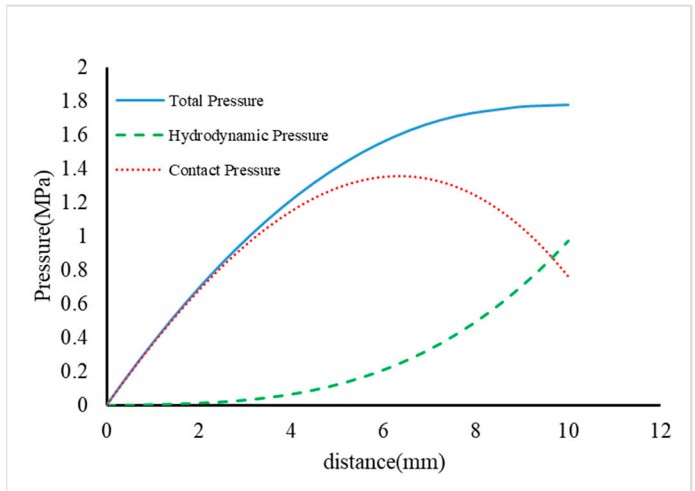

**Figure 10.** Pressure distribution on the surface with a roughness of 0.2 (μm).

The results obtained from Figure 10 show that the hydrodynamic pressure is highest in the vicinity of the outer diameter. The contact pressure reaches its highest value in the range of 5 to 8 (mm), while the applied load is 65 (N). The comparison of pressure distribution figures for rough surfaces shows that the contact pressure of surfaces with higher roughness is higher than that of those with lower roughness. This causes the hydrodynamic force of surfaces with higher roughness to decrease.

This section investigates the effect of force, viscosity, and rotation speed on the contact force. Figure 11 indicates the pressure distribution according to different forces.

The achieved results show that with the increase in force, the contact pressure in the area adjacent to the outer diameter increases. This indicates that with the increase in force, the contact area increases, and the slope of the graph also increases. A 30 (N) force creates the highest amount of contact pressure at a 6.2 (mm) distance. With increasing the force, the highest amount of contact pressure in the vicinity of the area of 5.8 (mm) is created.

Figure 12 shows the distribution of the contact pressure in terms of viscosity. According to the achieved results, it can be seen that the maximum contact pressure is created for the fluid with a viscosity of 0.65. The increase in viscosity creates a maximum contact pressure in the range of 5.2 (mm) up to 5.7 (mm). The increase in contact pressure due to the increase in viscosity can be seen in the lower penetration of the fluid in the contact area, which increases the contact force with the lower penetration of the fluid.

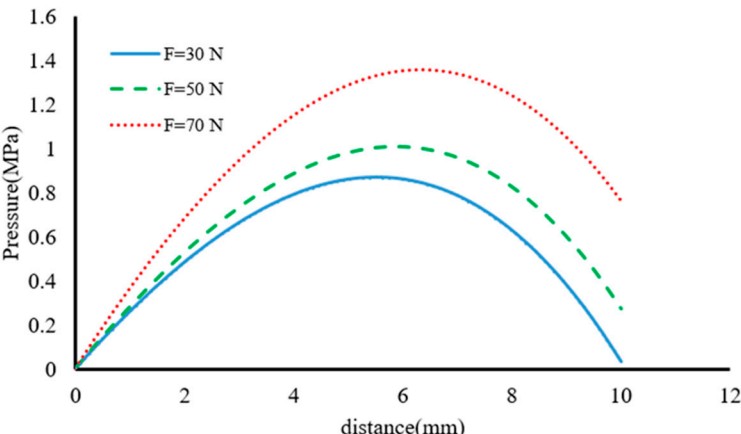

**Figure 11.** Pressure distribution according to the applied force.

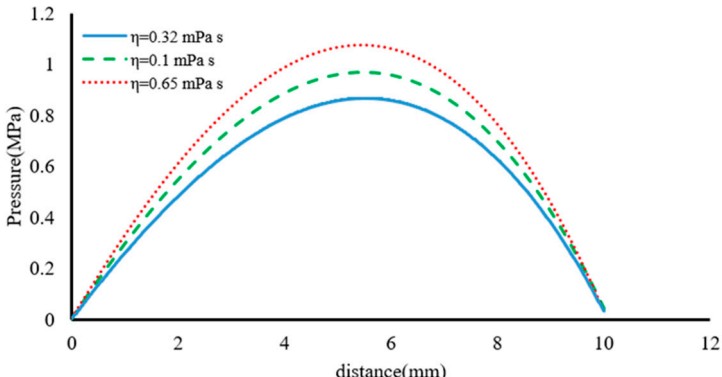

**Figure 12.** Contact pressure distribution in terms of viscosity.

Figure 13 shows the contact pressure distribution according to the rotation speed. Increasing the rotation speed causes an increase in the contact pressure in such a way that the contact pressure in the area adjacent to the outer diameter increases, which shows that the increase in the rotation speed leads to increases in the contact area.

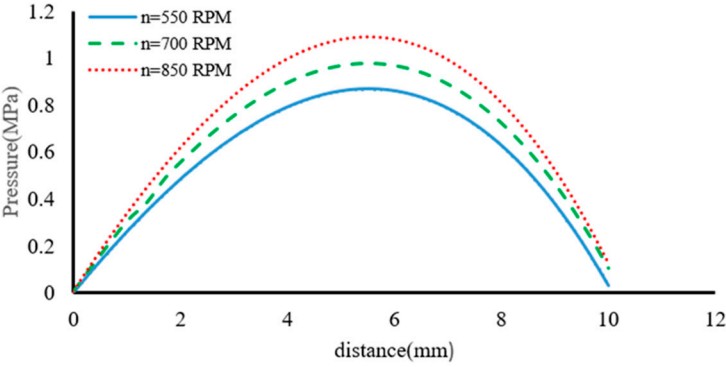

**Figure 13.** Contact pressure distribution according to rotation speed.

The effect of different parameters on the contact of two rough surfaces on the film thickness distribution has been investigated. Figure 14 shows the effect of force on the film thickness distribution.

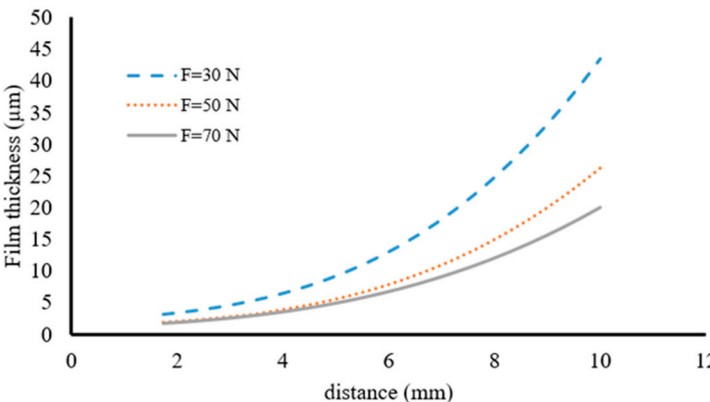

**Figure 14.** Effect of force on film thickness distribution for contact with rough surfaces.

According to the achieved results, it can be seen that there is a convergence of the film thickness in the vicinity of the inner diameter. This convergence is due to the application of the spring force, which results in a minimum value of fluid penetration in this area. Conversely, in the vicinity of the outer diameter, the effect of the force on the film thickness is quite clear. Specifically, it is observed that the film thickness decreases significantly with an increase in the applied force. For example, when comparing the film thickness with an applied force of 70 N to that with a force of 30 N, the former is approximately 50% of the latter.

According to Figure 15, an increase in surface roughness leads to a decrease in film thickness. The reason for this can be seen in preventing fluid penetration into the parts adjacent to the inner diameter. By reducing the roughness of the surface, the penetration of fluid to different parts between the two surfaces increases, increasing the film thickness.

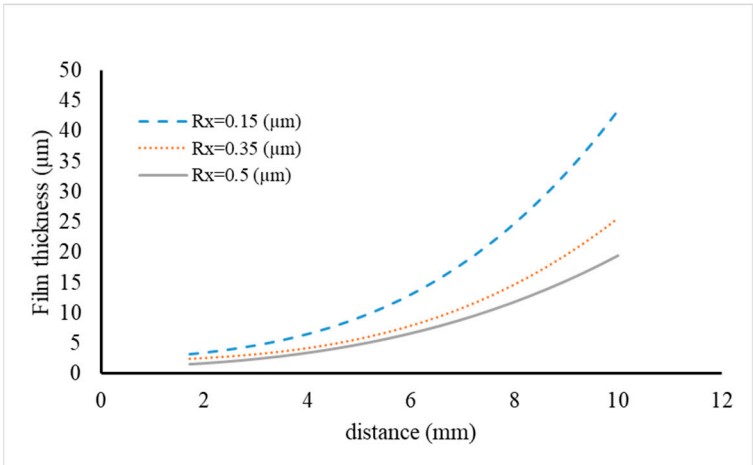

**Figure 15.** Effect of force on film thickness distribution for contact with rough surfaces.

Figure 16 shows the film thickness distribution due to fluid viscosity under the influence of surface roughness. The achieved results indicate that the film thickness decreases with increased viscosity. The reason can be a reduction in fluid penetration at a distance between two surfaces.

Due to this, the set of mechanical forces was considered as a whole, and the effect of the total force was investigated. According to the results achieved for the surfaces of the sealing components in the thermal condition, it can be concluded that the slope of the film thickness variation in the vicinity of the contact surface is low. As the distance from the contact surface increases, the slope of the variation increases. On the other hand, this condition is reversed for the pressure distribution.

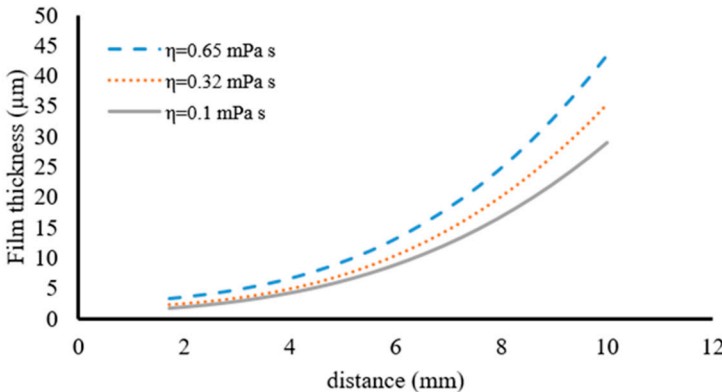

**Figure 16.** Effect of fluid viscosity on film thickness distribution for contact with rough surfaces.

The distribution of temperatures in the contact area is depicted in Figure 17. According to the temperature distribution, it can be observed that the highest temperature is reported in the vicinity of the inner diameter. This value reaches its minimum value in the vicinity of the outer diameter due to fluid penetration and lubrication properties.

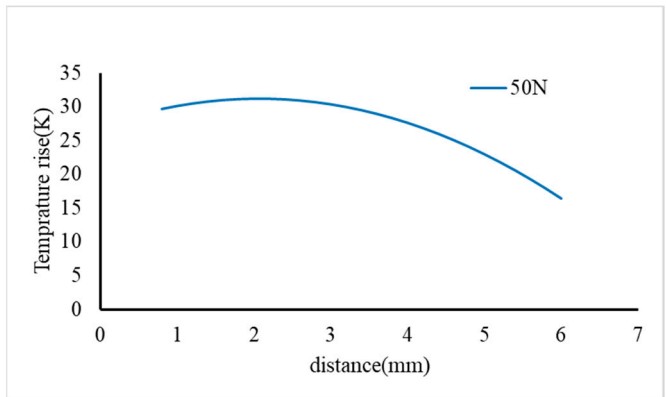

**Figure 17.** Temperature distribution in the contact area.

The distribution of fluid leakage with respect to various surface roughnesses is depicted in Figure 18. For this purpose, a fluid density of 933 $\frac{\text{Kgr}}{\text{m}^3}$ and a fluid viscosity of 6.34 mPa.s are considered. Validation of the results has been conducted according to reference [18].

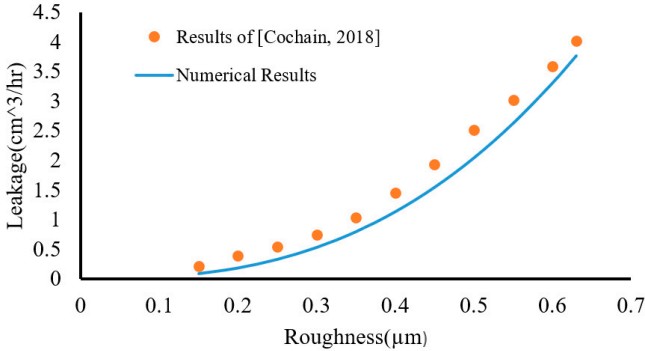

**Figure 18.** Fluid leakage distribution in relation to the different surface roughness [18].

The results obtained from Figure 16 show that fluid leakage increases as surface roughness increases. This can be attributed to increased surface roughness allowing more fluid to penetrate between asperities. Consequently, a larger volume of fluid escapes from between the asperities, leading to an overall increase in fluid leakage.

Friction is one of the most crucial variables when analyzing the contact effect. It is clear that the roughness of the contacting surfaces affects the performance of the tribo-system. However, the changes in the friction coefficient in relation to the rotational speed of the mechanical seal are studied in Figure 19, and its results have been compared with the previous study [36]. The surface roughness is 1 μm, the fluid viscosity is 0.035 Pa.s, and the applied pressure is 1 MPa. The graph of the results is as follows:

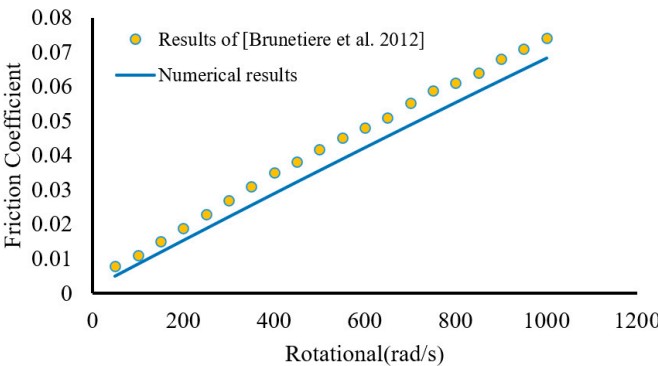

**Figure 19.** Validation of friction relative to surface roughness [36].

Figure 19 shows the change in friction coefficient according to rotational speed. As the speed increases, the contact between the asperities increases, the deformation of the asperities increases, and finally, the friction increases.

## 4. Conclusions

This paper investigates the performance of mechanical seals in the mixed lubrication regime using the load-sharing method. Based on this method, the load applied to the surfaces of the mechanical seal is carried by the fluid film and the surface roughness, and the load applied to the surface roughness results in asperity deformation. This research shows the effect of operating conditions such as applied load, speed, fluid viscosity, and surface roughness on the performance of the mechanical seal, as well as film thickness, hydrostatic pressure, and contact pressure.

An increase in the surface roughness leads to an increase in the contact between the asperities and an increase in the contact pressure. This causes a decrease in the film thickness, and as a result, the friction coefficient increases. Fluid film thickness, hydrodynamic pressure distribution, contact pressure distribution resulting from surface roughness, and friction are the outputs of this model.

The friction increases by 73% once the surface roughness increases from 0.3 (μm) to 0.5 (μm). Using a lubricant with a lower viscosity leads to a decrease in the film thickness. This increases the contribution of the contact pressure compared to the hydrodynamic pressure, which affects the friction coefficient. It should be mentioned that viscosity significantly affects the distribution of the contact pressure, while the maximum contact pressure is reported in the range of 5.2 (mm) up to 5.7 (mm). The contact pressure increases by 10.5% as the viscosity increases from 0.32 (MPa s) to 0.65 (MPa s). Lower fluid penetration in the contact region indicates increased contact pressure caused by increased viscosity, which raises the contact force due to the higher contact pressure.

The contact pressure rises as the shaft's rotational speed increases. Therefore, the friction simultaneously rises due to the increased contact pressure. Applying spring and screw forces to two fixed and moving parts results in permanent contact in this region. This is shown by the achieved results, which demonstrate that the variation in the fluid film thickness, contact force, and hydrodynamic force in the vicinity of the inner diameter is very small. In this paper, the modified Reynolds and deformation equations are solved for a set of points along the contact surface using the load-sharing method. The achieved results have been verified by comparing them to different references. Based on the achieved results,

it can be concluded that surface roughness changes significantly influence the performance of mechanical seals. Increasing the surface's roughness reduces the fluid film thickness and increases the contact force. The contact force increases as the force on the seal increases. However, the contact force increases as the force applied to the seal increases. Additionally, the contact force grows as the rotating speed increases.

The novel aspect of this study is the simultaneous solution of the modified Reynolds equations, deformation equations, and contact equations to predict the performance of the mechanical seal using the load-sharing method. The findings from the load-sharing concept are legitimate, and the error percentage is quite low, according to a comparison between the results from the articles and the results from the load-sharing concept.

**Author Contributions:** Conceptualization, M.R. and S.A.; methodology, M.R. and S.A.; software, M.R.; validation, M.R.; formal analysis, A.S. and S.A.; investigation, M.R. and S.A.; resources, M.R.; data curation, M.R.; writing—original draft preparation, A.S.; writing—review and editing, A.S.; visualization, A.S.; supervision, S.A.; project administration, S.A. All authors have read and agreed to the published version of the manuscript.

**Funding:** This research received no external funding.

**Data Availability Statement:** The data that supports the findings of this paper are available from the corresponding author.

**Conflicts of Interest:** The authors declare no conflict of interest.

### Nomenclature

| | | | |
|---|---|---|---|
| $A_i$ | Surface region | $u_y$ | Velocity in the y direction |
| $A_t$ | Overall actual contact surface | $\bar{V}$ | Equivalent velocity in the y direction |
| $C_p$ | Specific heat capacity at constant pressure | $v$ | Elastic deformation |
| $d$ | Average surface distance of two surfaces | $W_i$ | Load applied by the asperity |
| $D$ | Mechanical seal diameter | $w_t$ | Total load |
| $E'$ | Effective elasticity coefficient | $y_s$ | The average distance between the height line and the asperities' peak line |
| $G$ | Hardness factor | $Z_a$ | The height of an asperity before contact |
| $\bar{h}_t$ | Dimensionless local film thickness | $\bar{U}$ | Equivalent velocity in the x direction |
| $h$ | Fluid film thickness | $\alpha$ | Viscosity-pressure coefficient |
| $h_0$ | The initial thickness of the fluid film | $\beta$ | Roughness radius |
| $Hd$ | Vickers hardness of softer material | $\acute{\delta}$ | Elastic deformation |
| $K$ | Surface thermal conductivity coefficient | $\delta_1$ | Roughness height |
| $N_a$ | Number of asperities on the surface | $\delta_i$ | Asperity height |
| $n_a$ | Surface density of asperities | $\delta'$ | Roughness Deformation |
| $P$ | Total pressure | $\xi$ | Overall interference |
| $p_d$ | Contact pressure | $\eta$ | Fluid viscosity |
| $P_f$ | Hydrodynamic pressure | $\rho$ | Fluid density |
| $p_m$ | Average contact pressure | $\sigma$ | Surface roughness distribution |
| $R_x$ | Roughness radius | $\sigma_s$ | Peak asperities' standard deviation |
| $t$ | Time | $\phi(z)$ | Distribution density of the height of asperities |
| $T$ | Temperature | $\phi_i$ | Pressure flow coefficient |
| $u_m$ | Average displacement of the asperity height | | |
| $u_g$ | Deformation caused by pressure | | |
| $\bar{U}$ | Equivalent velocity in the x direction | | |
| $u_s$ | Sliding speed | | |
| $u_x$ | Velocity in the x direction | | |

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
