# Peer review of "Application of Load-Sharing Concept to Mechanical Seals"

_lubricants, doi:10.3390/lubricants11060266_

Round 1

Reviewer 1 Report

The paper is within the scope of the journal. 

1- The Abstract is too long and needs to be rewritten to make is more concise and focused. It should state: (a)- the motivation behind the paper or the importance for such a study, (b)- what is new/original in this paper (see point 2 below), and (c )- what is the main finding of this paper.

2- Before a claim is made in the Abstract regarding the original contribution of the paper, it is important to note what is not already covered by other contributions included in the survey of literature in the paper. This means that the survey should include other papers, not included. In particular, the following should have been cited, which provide solution of seals under mixed lubrication, also including CFD:

Yin, T., Wei, D., Wang, T., Fu, J. and Xie, Z., "Mixed-lubrication mechanism considering thermal effect on high-pressure to reciprocating water seal", Tribology International, 2022, 175:107856.

Cheng, D., Gu, L. and Sun, Y., "Mixed Lubrication Modeling of Multi-Lip Reciprocating Seals Based on Elastohydrodynamic Lubrication Theory", Machines, 2022, 10(6):483.

Fatourehchi, E., Shahmohamadi, H., Rahmani, R., Rahnejat, H., Johnson, M. and Wilson, I., "Tribology of dust‐stop seals of mixing machines", Lubrication Science, 2023, 35(3):193-206.

Now with the inclusion of the above up to date references in the review of literature, the authors need to make the necessary claim to originality in the Abstract.

3- The last paragraph in the Introduction should expand on the originality of the paper.

4- There should be a full nomenclature of all the mathematical symbols used in the paper in alphabetic order at the end of the paper.

5- Please provide justification for use of a parabolic film shape in equation (4). Is this appropriate for flat seal surfaces? 

6- flow factors should be stated? I presume these are based on Patir and Cheng approach. If so, Patir and Cheng should be cited.

7- Do the authors realize that flow factors based on Patir and Cheng are for idealized rough surfaces with Gaussian distribution of asperity heights. This is not the case for seals, which are non-Gaussian rough surfaces. This point needs to be stated as an assumption made, see:

Nomikos, P., Rahmani, R., Morris, N. and Rahnejat, H., "An investigation of oil leakage from automotive driveshaft radial lip seals", Proceedings of the Institution of Mechanical Engineers, Part D: Journal of Automobile Engineering, 2022 : 09544070221127105.

8- The papers conclusions are rather well-known and are not focused. The Conclusions should be rewritten and in a bullet form with specific findings, the main one of which should have been also noted briefly in the Abstract. 

9- Paper's grammar needs improvement to some extent. Please break up the long sentences to shorter and more focused ones. In particular, the Introduction is rather too wordy, and not concise. 

10- friction is a force, so no need for "friction force".

11- There is no such term as elastioplastic. You obviously mean "elastoplastic".       

I look forward to receiving any revised version, with all the changes made clearly highlighted. Please also respond to all the points raised above in a point-by-point basis.

English grammar needs moderate improvement.

The paper is quite good, but authors need to address the issues raised.

Author Response

The authors would like to thank the reviewer for the valuable comments. We have tried our best to respond to all of the comments and revise the manuscript based on the comments.

Reviewer 2 Report

The paper deals with mixed lubrication regime in face mechanical seals. Although the presented calculation model implies the calculation of the temperature mode. However, why is the temperature distribution not presented in the paper? There is also no leakage calculation, a significant parameter for contact and non-contact mechanical seals. Verifying the obtained results with the results known from the literature is not entirely clear. Why is the film thickness distribution compared with a non-contact wavy-tilt-dam mechanical seal [35] (Fig. 6), which does not consider the effect of surface roughness? And the contact pressure distribution (Fig. 7) is generally taken for the O-ring [37] (Fig. 7)? Figure 7 shows the total pressure distribution, but the paper [36] presents the pressure in the film. Also, the boundary conditions in this paper (applied pressure 4 MPa and rotation speed 600 rpm) and the literature [36] (applied pressure 5x105 Pa, rotation speed 600 rad/s) do not match etc. Also, it seems that the magnitude of the applied force F does not correspond to the contact pressure distribution multiplied by the surface area of the ring (Fig. 9)? Why is friction force validation generally given for reciprocating hydraulic seals?

Because of the remarks made, this paper cannot be published in the Lubricants journal in its present form.

Author Response

(The authors gave the same response as above.)

Round 2

Reviewer 1 Report

The authors have adequately addressed all my concerns. I'm happy to recommend the revised paper for publication in Lubricants.

The quality of English grammar is mostly fine. May require some moderate editing at the proofs-stage.  

Author Response

We would like to thank the reviewer for his/her valuable comments.

Reviewer 2 Report

After the revision of the paper, some aspects have been improved, but there are still critical unanswered questions:

1. Unclear distribution of contact pressure on the contact surface of the mechanical seal. Why is the maximum contact pressure located close to the center of the contact surface of the seal (Fig. 8)? Although judging by the film thickness distribution (Fig. 6, 12), a gap (confusor shape) is present on the entire surface of the seal, on the outer diameter it is tens of micrometers, and on the inner diameter it is micrometers. How to even consider that the contact occurs on the inner diameter, then the maximum contact pressure should be in this place. The shape of the contact pressure distribution rather corresponds to the contact of a cylindrical surface with a flat one.

2. Why does the graph of hydrodynamic pressure (Fig. 8) have a concave shape? Does this correspond to the diffusor gap, not the confusor one? Why is the distribution of the total pressure on the outer surface of the ring not distributed? How is it created? Which is the load factor of a mechanical seal?

3. Why is the contact surface of a mechanical seal considered linear contact eq. (4)? What is the manifestation of hydrodynamic lubrication if fluid pressure distribution is more similar to hydrostatic pressure distribution?

4. With this distribution of total and contact pressures (Fig. 8), the applied load should be significantly greater than indicated in Fig. 9,12. What is the value of the contact surface area and the outer and inner radii of the mechanical seal?

5. Why does the text of the paper state that the isothermal condition is used (Page 14, line 392)?

6. Why are there different symbols of the roughness of Rx (Fig. 6) і Sq (Fig. 13)?

7. What is the unit of Contact Load in Figure 7?

8. Where in equation (3) symbol ∅х is used?

9. It is necessary to check for text and references errors. For example, on page 13, line 367, there should be Figure 12.

Author Response

We would like to appreciate the reviewer for the valuable comments. Please see the attached file.

Round 3

Reviewer 2 Report

Even after the revision of the paper and the corrections made, I do not quite understand the authors' explanations regarding the contact and hydrodynamic pressure distributions and the reason for not covering the load factor of the mechanical seal in the paper. Why is there no equation for the balance of forces (closing and opening forces) acting on the moving ring?
